# Desiccant Films Made of Low-Density Polyethylene with Dispersed Silica Gel—Water Vapor Absorption, Permeability (H_2_O, N_2_, O_2_, CO_2_), and Mechanical Properties

**DOI:** 10.3390/ma12142304

**Published:** 2019-07-18

**Authors:** Sven Sängerlaub, Esra Kucukpinar, Kajetan Müller

**Affiliations:** 1Fraunhofer Institute for Process Engineering and Packaging IVV, Giggenhauser Strasse 35, 85354 Freising, Germany; 2TUM School of Life Sciences Weihenstephan, Chair of Food Packaging Technology, Technical University of Munich, Weihenstephaner Steig 22, 85354 Freising, Germany; 3Faculty of Mechanical Engineering, University of Applied Science Kempten, Bahnhofstraße 61, 87435 Kempten, Germany

**Keywords:** active packaging, moisture control, water vapor absorption, effective diffusion coefficient, mixed matrix membranes

## Abstract

Silica gel is a well-known desiccant. Through dispersion of silica gel in a polymer, films can be made that absorb and desorb water vapor. The water vapor absorption becomes reversible by exposing such films to a water vapor pressure below that of the water vapor pressure during absorption, or by heating the film. The intention of this study was to achieve a better understanding about the water vapor absorption, permeability (H_2_O, N_2_, O_2_, CO_2_), and mechanical properties of films with dispersed silica gel. Low-density polyethylene (PE-LD) monolayer films with a nominal silica gel concentration of 0.2, 0.4, and 0.6 g dispersed silica gel per 1 g film (PE-LD) were prepared and they absorbed up to 0.08 g water vapor per 1 g of film. The water vapor absorption as a function of time was described by using effective diffusion coefficients. The steady state (effective) water vapor permeation coefficients of the films with dispersed silica gel were a factor of 2 to 14 (8.4 to 60.2·10^−12^ mg·cm·(cm^2^·s·Pa)^−1^, 23 °C) higher than for pure PE-LD films (4.3·10^−12^ mg·cm·(cm²·s·Pa)^−1^, 23 °C). On the other hand, the steady state gas permeabilities for N_2_, O_2_, and CO_2_ were reduced to around one-third of the pure PE-LD films. An important result is that (effective) water vapor permeation coefficients calculated from results of sorption and measured by permeation experiments yielded similar values. It has been found that it is possible to describe the sorption and diffusion behavior of water by knowing the permeability coefficient and the sorption capacity of the film ($P_{\mathrm{eff}.}\approx S_{\mathrm{eff}.}\cdot D_{\mathrm{eff}.}$). The tensile stress changed only slightly (values between 10 and 14 N mm^−2^), while the tensile strain at break was reduced with higher nominal silica gel concentration from 318 length-% (pure PE-LD film) to 5 length-% (PE-LD with 0.6 g dispersed silica gel per 1 g film).

## 1. Introduction

It is a well-known fact that the water activity of foods and pharmaceuticals has an influence on their shelf-life and stability [1,2,3,4,5,6,7,8,9,10,11,12,13]. Water vapor sensitive pharmaceuticals are protected by the use of desiccants [11]. Growth of microorganisms, powder-agglomeration [2,10,14,15,16,17,18,19,20,21], and loss of crispness of snack products [2,8,22,23] are examples of food deterioration caused or enabled by a too high water vapor pressure in their direct surrounding, or by a too high water activity of these products. For these reasons, the application of desiccants in packages is relevant with an aim of binding the excess water. Desiccants can be filled into sachets, which are added to packaging. Another option is the dispersion of a desiccant powder in a polymer matrix and to form packaging materials with these compounds, such as films, by film-extrusion. Dispersing desiccants and water vapor absorbers in polymer structures is a strategy described already by other research groups. Materials made of desiccants and water vapor absorbers such as calcium chloride, sodium chloride, calcium oxide, magnesium oxide, zeolites and molecular sieves, silica gel, and metal-organic frameworks dispersed in various polymers have already been tested [24,25,26,27,28,29,30,31,32,33,34,35,36,37,38,39,40,41,42,43,44,45,46].

Silica gel is a nanoporous material which adsorbs and desorbs water vapor. The water vapor adsorption increases with increasing relative humidity (RH) [47]. In silica gel with pore diameters of a few nanometers, capillary condensation occurs. The smaller the pore diameter, the smaller the water vapor pressure at which water vapor condensation in the capillaries occurs [48]. In substances with a low pore diameter, water vapor condenses at a RH of a few percentages [7,49,50]. As an example, molecular sieve 5Å with a pore diameter of 0.43 nm absorbs water vapor down to a few percent RH [49]. The water vapor sorption isotherm has, at low RH, a sharp increase of water vapor absorption. On the other hand, in silica gel and molecular sieves with higher pore diameters, water vapor condenses at higher levels of RH [49,50,51] of >50% RH. Silica gel has a pore size distribution. Therefore, capillary condensation occurs at a broad range of RH. The water vapor sorption isotherm can be described by the dual-sorption model. The water vapor adsorption capacity at 80% RH, at 23 °C, is 0.3 to 0.4 g water per 1 g of silica gel [47,52]. The water vapor sorption isotherms of silica gel are dependent on the kind of silica gel, and therefore, can differ. As the temperature increases, the adsorbed amount of water decreases at a constant RH [53].

The main intention of our study was to analyze and describe the water vapor absorption, the gas and vapor permeability (H_2_O, N_2_, O_2_, CO_2_), and the mechanical properties of films with dispersed silica gel. Our hypothesis was that the effective permeation and the effective diffusion coefficients determined by water vapor sorption and water vapor permeation experiments yield similar results, which has not been analyzed before. Furthermore, we assumed that the sorption behavior of low-density polyethylene (PE-LD) films with dispersed silica gel can be described in a simplified manner by equations derived from the diffusion–solution model. Silica gel was chosen because it absorbs water vapor reversibly, and therefore, the concept of an “effective sorption coefficient”, which implicates absorption and desorption, for polymer blends made thereof can be applied (see Equation (6)). Besides, silica gel is solid and can be well dispersed in polymers. A further intention was the characterization of gas permeability and mechanical properties of these films.

It is worth mentioning that the authors of this study analyzed other desiccant films made of low-density polyethylene with dispersed calcium oxide in a former study [46]. One important difference between both substances is that silica gel binds water molecules by physisorption, which is reversible, and calcium oxide by chemisorption, which is irreversible.

## 2. Material and Methods

### 2.1. Materials

#### 2.1.1. Silica Gel

Silica gel beads with a diameter of two to five millimeters (Merck, order number 107735, Darmstadt, Germany) were used as desiccant [54]. The silica gel was milled with a ball mill (S 1000, Retsch, Haan, Germany) for 60 min with 10 balls (5 balls with a diameter of 13 mm and 5 balls with a diameter of 17 mm) made of sintered aluminum oxide.

The density of the silica gel was estimated in order to calculate the volume fraction of dispersed silica gel in the polymer (Equation (1)). Silica gel consists of silicon dioxide and pores. The pores are filled with air in dry conditions. We estimated the density of silica gel from the density of silicon dioxide and the measured values for the water vapor absorption at saturation. We made the assumption that the pores are fully saturated and filled with water vapor at high RH with a density of the adsorbed water of 1 g/cm^3^. Silica gel adsorbed at 91% RH 0.32 g water/g dry silica gel. When linearly extrapolated to 100% RH the water vapor absorption is 0.35 g water/g of dry silica gel which equals to a pore volume of 0.35 cm^3^/g of dry silica gel. This value is within the range of literature values, with 0.40 to 0.43 cm^3^ pore volume/g of dry silica gel [47]. Silicon dioxide has a density of 2.2 g/cm^3^ [47,49,55,56] and has therefore a specific volume of 0.45 cm^3^/g. The specific volume is the sum of the volume of the pores (0.35 cm^3^/g) and the volume of silicon dioxide (0.45 cm^3^/g), i.e., 0.80 cm^3^/g. The density of the silica gel calculates, according to this estimation, as 1.243 g/cm^3^. This value is in the range of literature values, which vary between 0.75 to 1.25 g/cm^3^ [49]. The porosity of silica gel calculates to 0.44. 

#### 2.1.2. Polyethylene

The polymer matrix was a low-density polyethylene (PE-LD, Lupolen 1806 H, Basell, Frankfurt a. Main, Germany) with a density of 0.92 g/cm^3^ and a melting temperature of 108 °C [57]. 

### 2.2. Methods

The authors of this study analyzed other desiccant films with calcium oxide before [46]. The description in the Section 2.2.1, Section 2.2.2, Section 2.2.4, Section 2.2.5, Section 2.2.7, and Section 2.2.8 are in some parts similar. To improve the readability of our publication these methods are described here again, however, the descriptions were adapted to the materials used in this study.

#### 2.2.1. Film Production

The LD-PE pellets were dry blended with concentrations of 0.2, 0.4, and 0.6 g silica gel/g silica gel powder blend. The blends were compounded using a parallel twin screw extruder (Rheomex PTW 16/25, ThermoElectron GmbH; extruder drive Rheocord300pm, Thermo Electron GmbH, Karlsruhe, Germany) with a circular die (diameter: Three millimeters) at temperatures between 150 to 210 °C. The melt strand was dry cooled in a closed container with dried silica gel as desiccant and converted into pellets with a pelletizer (SGS 50-E, CF Scheer & CIE, Germany). The compounds were then extruded again; in this second step, the compound was extruded to stripes using the same parallel twin screw extruder with a ribbon die (width 50 mm) at temperatures between 150 to 210 °C. The blend was extruded two times in order to better disperse silica gel in the polymer. Reference stripes of pure PE-LD without silica gel were produced in the same way. The extruder screw was equipped with shear elements. The same screw configuration as in a previous publication of ours was taken (see Figure 1 in [46]). The screw diameter was 15.6 mm, the screw length 400 mm (L/D = 25).

For the measurement of the (effective) water vapor permeability coefficients (WVPC), the films were pressed for 5 s on a heated platen press (341-50-12×12, Loomis products Kahlefeld GmbH, Kahlefeld, Kaiserslautern, Germany) at a temperature of 130 °C and a force of 30 kN, to increase the sample area to 15 cm × 15 cm. In Table 1 the thicknesses of the samples, measured with a calliper (Solar-Absolute Digimatic Calliper, Mitutoyo, Kawasaki, Japan), are shown.

For the determination of water vapor permeability coefficients, samples were pressed to increase the area up to around 10 cm × 10 cm for the permeation testing.

#### 2.2.2. Density of Films and Concentration of Silica Gel in Films

The density of the stripes was measured with a density measurement kit (Mettler Toledo, Gießen, Germany). The mass fraction of silica gel *x* can be determined from the density of silica gel *ρ*_silica gel_, the density of the polymer *ρ*_polymer_, and the density of films with dispersed silica gel *ρ*_blend_, according to Equation (1). The mass fraction takes values between zero (no filler) and one (only filler). 

(1)$$x=\frac{\rho_{silica\;gel}\times \rho_{polymer}-\rho_{silica\;gel}\times \rho_{blend}}{\rho_{blend}\times \left(\rho_{polymer}-\rho_{silica\;gel}\right)}$$

#### 2.2.3. Electron Microscopic Pictures

Electron microscopic pictures were taken with a field emission scanning electron microscope (SEM S-7000, Hitachi Ltd., Tokio, Japan) at 500- and 1000-× magnification, with a nominal resolution of 15 nm. From the film samples, sections with a thickness of 20 to 50 µm were taken with a rotation microtome. In high vacuum, the samples were coated with a gold layer at a thickness of 5 nm. 

#### 2.2.4. Water Vapor Absorption

The weight of the samples was measured with laboratory scales (Mettler Toledo, Gießen, Germany, type Delta Range AT 261, and Sartorius AG, Göttingen, Germany, type 1702 MP8). The mass fraction of absorbed water is presented in relation to the dry mass of silica gel or of the films.

The measurements were carried out at defined RHs and a temperature of 23 °C. The samples were stored in desiccators with saturated solutions of potassium hydroxide (KOH), sodium hydrogen sulphate (NaHSO_4_), and potassium nitrate (KNO_3_), with residuum at the bottom of the desiccators. The equilibrium RH (ERH) levels reported by Greenspan were 9% for KOH, 52% for NaHSO_4_, and 92% for KNO_3_ [58]. The desiccators were equipped with ventilators to ensure a uniform RH inside the desiccators. The measured ERH levels inside the desiccators were 9.3% ± 1.5% for KOH, 52.0% ± 1.5% for NaHSO_4_, and 91.4% ± 1.5% for KNO_3_.

For the measurement of the water vapor absorption of silica gel, around 1 g silica gel powder was spread to a thin film on a watch glass. The watch glasses were stored in desiccators with the RH levels as described above. For one measurement point, five specimens were characterized, and an average was taken.

The water vapor absorption of the stripes was measured using squares taken from the middle of the stripes, squares with a side length of 3 cm. For one measurement point, five specimens were characterized, and an average was taken.

The time dependent water vapor absorption of films from two sides is described by Equation (2) [59]. Amount of water absorbed within the time interval [0, *t*] is represented by *m_t_*. *m_∞_* represents the amount of absorbed water in equilibrium, i.e., *t* → ∞. *l* represents the thickness of the samples, *m_t_/m_∞_* represents the relative mass fraction of water in the film, and *D* is the diffusion coefficient.

(2)$$\frac{m_{t}}{m_{\infty }}=1-\overset{\infty }{\underset{n=0}{\sum }}\frac{8}{(2n+1{)}^{2}\times \pi^{2}}\times e^{\frac{-D\times (2n+1{)}^{2}\times \pi^{2}\times t}{l^{2}}}$$

For short periods, Equation (2) simplifies to Equation (3) [60,61],

(3)$$\frac{m_{t}}{m_{\infty }}=\frac{4}{\sqrt{\pi }}\times \sqrt{\frac{D\times t}{l^{2}}}$$

The mass fraction *m_t_/m_∞_* as a function of the square root of time, $\sqrt{t,}$ is approximately linear up to a value of *m_t_/m_∞_* that is below 0.5 to 0.7 [60,62,63,64,65,66,67]. In this work, Equation (3) is applied for a mass fraction of absorbed water vapor with *m_t_/m_∞_* being less than 0.7.

Strictly speaking, Equations (2) and (3) are only valid if the sorption follows the solution–diffusion model [59,68,69], and if Henry’s law is valid [69]. The application of these equations to the here considered materials is a simplification, and therefore, *D* should be interpreted as an effective diffusion coefficient *D*_eff._ [36].

From the amount of absorbed water in equilibrium, the effective sorption coefficient was calculated assuming in a simplified manner Henry’s law [68,69,70,71,72,73,74,75,76,77]. In this simplified case, the concentration of water $c_{water}$ in the silica gel is linearly dependent on the water vapor partial pressure $p_{water}$ (Equation (4)). The proportionality constant is the sorption coefficient S and again it should be interpreted as effective solubility coefficient, S_eff.,_ in this work:(4)$$c_{water}=S\times p_{water}$$

#### 2.2.5. Water Vapor Permeation

The water vapor permeation coefficient (WVPC) of the films was measured by applying an electrolytic method (DIN EN ISO 15106-3) using a permeation-testing device type AQUATRAN (Mocon Inc., Minneapolis, MN, USA). The gradient of RH between both sides of the sample was set to 85% to 0% RH. The measurement temperature was 23 °C. All film samples were thermo-pressed before the measurement to reduce their thickness (see Table 1) and to increase the area to around 10 cm × 10 cm required for the testing device. The film samples with dispersed silica gel were stored at 0% RH before measurement.

With water vapor permeation experiments in transient state, the diffusion coefficients can be determined. The lag-time, θ, according to *Barrer*, is the intercept at the time-axis at which the tangent, at the permeated amount of substance in stationary state, intersects the time-axis (Equation (5)) [59,74,78]. l is the thickness and D is the diffusion coefficient. After 2.7-times of the lag-time, the flux density of the diffusing substance differs only marginally from the stationary value of the flux density [59,79].

(5)$$\theta =\;\frac{l^{2}}{6\times D}$$

The effective permeation coefficient, $P_{eff.}$ can be calculated according to Equation (6) [59]. It is the product of the effective sorption coefficient, $S_{eff.}\;$, which can be calculated according to Equation (4), and the effective diffusion coefficient $D_{eff.},$ which can be calculated by Equation (5) [59].

(6)$$P_{eff.}\approx S_{eff.}\times D_{eff.};P_{eff.}\approx constant$$

#### 2.2.6. Nitrogen, Carbon Dioxide, and Oxygen Permeability

The gas permeability (manometric method) was measured according to DIN 53 380-2 with a permeation testing device (GDP-E, Brugger Feinmechanik GmbH, Munich, Germany) with the dry gases, nitrogen, carbon dioxide, and oxygen. The test samples were dried before the measurements.

#### 2.2.7. Mechanical Tests and Elongation

Mechanical tests were performed according to DIN EN ISO 527-1 and DIN EN ISO 527-3, using a tensile testing machine (model 3962, Schenck Trebel, Germany). The elongation (see Appendix A) was measured with a calliper (Solar-Absolute Digimatic Calliper, Mitutoyo, Kawasaki, Japan). For the measurement of the elongation, the same samples were taken as used for the water vapor absorption measurements. 

#### 2.2.8. Confidence Intervals

All values are presented as average values (arithmetic mean). The error bars given in the following figures (Results and Discussion section) represent the confidence intervals with a significance level of 95% [80]. Absorption measurements were done by using five specimens, mechanical tests were performed using ten specimens, and WVPC measurements were performed using two specimens for each measurement.

## 3. Results and Discussion

### 3.1. Silica Gel Concentration in the Films

The density of the films were measured to calculate the sorption coefficient of the filled polymers (as discussed in Section 3.4) and to estimate the real concentration of silica gel in the films. In Table 2, the measured and expected densities of films with dispersed silica gel and of polyethylene and silica gel are presented (see Equation (1)). By the results, it is shown that the measured densities of the filled films were higher than the expected densities. For the film with 0.6 g silica gel/g film, the measured film density (1.35 cm^3^/g) was higher than the density of the pure silica gel (1.24 cm^3^/g). A possible reason for this difference between measured and expected value was the penetration of polymer molecules into the silica gel matrix and the compression of the silica gel by the pressure during extrusion. However, our results are not sufficient to prove these assumptions, and the reasons for the difference are open to further investigation. Due to the fact that these possible effects could not be evaluated and proved, it was also not possible to calculate the real silicon oxide concentration in the films by using Equation (1). For this reason, the nominal silica gel concentrations (0.2, 0.4, and 0.6 g silica gel/g film) are shown in this work and are used for further calculations.

### 3.2. Microscopic Pictures

In Figure 1 and Figure 2 the electron microscopic pictures of films with dispersed silica gel particles are shown. In the Appendix A, additional electron microscopic figures are presented. Through the milling of silica gel, particle diameters much lower than the thickness of the analyzed film were gained. The bigger particles in the figures had a diameter of 30 µm to 50 µm. Their diameter was much smaller than the thickness of the analyzed films, which vary from 230 µm to 1870 µm. The films with 0.2 g silica gel/g film (Figure 1) had the silica gel particles fully embedded in the polymer matrix. It can be assumed from this observation that the diffusion of water vapor in the films was primarily influenced by the polymer matrix (solution–diffusion process) in these samples. The film with 0.6 g silica gel/g film (Figure 2) has agglomerated silica gel particles. In this structure, transport mechanisms other than the solution–diffusion process can occur, such as the diffusion at the particle surface and through the channels between the adjacent particles. This aspect is further discussed in Section 3.5.

### 3.3. Effective Diffusion Coefficients

The effective diffusion coefficients were determined by two methods: Sorption and permeation experiments, both in transient state.

#### 3.3.1. Effective Diffusion Coefficients by Sorption Experiments

To determine $D_{eff.}$ from sorption experiments according to Equation (3), the relative mass fraction (*m_t_/m_∞_*) of sorbed water vapor as a function of the square root of time divided by the thickness for the measurements at 9% RH, 52% RH, and 91% RH was depicted in Figure 3, Figure 4 and Figure 5, respectively. The water vapor sorption was approximately linear at a mass fraction between 0 and 0.7. From these linear sections, the effective diffusion coefficients were calculated. By the curves of the films with 0.6 g silica gel/g film a non-linear behavior is shown starting at *m_t_/m_∞_* values above 0.7, indicating a pseudo-Fick’s absorption behavior that occurs at accelerated diffusions. Pseudo-Fick’s absorption is described phenomenologically by other research groups with the dual-stage model [82,83,84,85]. It is based on the superposition of two sorption processes, e.g., two dual-stage sorption processes, which can be described by two diffusion coefficients and two sorption coefficients. A reason for this is slow stress relaxation by the sorbent during, or after, the first sorption process, which causes further sorption by the sorbent [83,86].

In Table 3, the calculated effective diffusion coefficients are shown. The experimentally determined effective diffusion coefficients of the films with 0.2 and 0.4 g silica gel/g films were found to be between 3.1 and 6.1 cm^2^/s for all tested RH conditions. Through dispersion of silica gel in PE-LD, the effective diffusion coefficient of the filled polymer matrix compared to pure PE-LD was reduced. The effective diffusion coefficients of the films with 0.6 g silica gel/g film were up to a factor of three higher than the other filled films. With increasing RH, the effective diffusion coefficients increase. This effect was more pronounced for the case of the 0.6 g silica gel/g film than the other films. At high silica gel concentrations, the silica gel particles touched each other. For this reason, further diffusion mechanisms besides the solution–diffusion mechanism are possible, such as the diffusion on the surface of touching particles or capillary condensation between the touching particles and between the polymer wall [87,88,89,90]. A higher silica gel concentration, therefore, increases the inner surface which is available for surface diffusion and capillary condensation. Both diffusions mechanisms have a stronger effect at higher silica gel concentration and higher RH.

The effective diffusion coefficients of the filled films were by two orders of magnitude smaller than that of the polyethylene films. This contradiction is further discussed in Section 3.5.

In Figure 6, Figure 7 and Figure 8, the calculated and measured water vapor sorption of the films are shown as a function of time. The symbols are the measured values; the lines are the calculated values according to Equation (2) using the *D_eff._* values shown in Table 3. The calculated curves represent Fick’s absorption. The deviations at 91% RH could be explained by pseudo-Fick’s absorption behavior, as discussed before. The water vapor sorption of the films could be described by Equation (2) with adequate accuracy. Therefore, it can be concluded that the diffusion within the polymer matrix was the limiting step for the absorption of the films and not the adsorption rate of the silica gel, which is a pre-condition for the application of the equations derived by Fick’s law. For the application of effective coefficients of filled polymers, it was decisive that the filled polymers could be described as quasi-homogeneous materials. It needs to be mentioned here that the used equations derived from Fick’s laws are only valid for the pure polymer matrix between the particles. This is of relevance for the estimation of the effective diffusion coefficient from the effective permeation coefficient of the filled film as also discussed in Section 3.5. The water vapor sorption rate of the filled films was determined using the water vapor permeation coefficients of the polymer matrix embedding the silica gel particles.

#### 3.3.2. Effective Diffusion Coefficients by Permeation Experiments

The lag-times for water vapor permeation were experimentally determined. From these lag times, the effective diffusion coefficients were calculated according to Barrer (Equation (5)) [69,74]. The calculated diffusion coefficients are shown in Table 3, the graphs are in Appendix A. As expected, the lag time of pure PE-LD films was shorter than that of the films with silica gel. By the activity of an absorber, the lag time increased [92,93]. It is reported that the higher the absorber concentration, the higher the effective lag time becomes, i.e., the effective diffusion coefficient gets smaller [92,94,95,96]. However, the film with 0.6 g silica gel/g film had the highest effective diffusion coefficient. The reason was the existence of channels in the polymer matrix, as discussed before. Besides this film with the highest silica gel concentration, the effective diffusion coefficient was reduced by silica gel because silica gel absorbed water vapor directly after the start of the permeation experiment.

Even before the lag time was reached for the polymer-blends, the water molecules permeated through the polymer. The curve for the permeated amount of water through a film as a function of time became more steep (less or no water vapor permeation before lag time), the higher the reaction rate [96,97,98,99,100] and the lower the particle size [98], i.e., a sharply defined progressing reaction front forms in the polymer matrix.

The effective diffusion coefficients, *D_eff._*, calculated from the permeation measurements had a range of variation of a factor between 0.6 to 2.4 in relation to the effective diffusion coefficients obtained from the sorption measurements, which is within the typical range for such measurements. By these results is shown that sorption and permeation measurements yield similar results for *D_eff._*. Therefore, we conclude that the diffusion behavior of films with dispersed silica gel can be formally described by the solution–diffusion model.

### 3.4. Water Vapor Absorption Capacity and Effective Sorption Coefficients

The effective sorption coefficients, $S_{eff.}$, were calculated (Table 4) using the mass fraction of water in the films (water vapor sorption capacity; Figure 6, Figure 7 and Figure 8), the density of the films (Table 2), and the water vapor saturation pressure of 28.08 mbar (2808 Pa) at 23 °C [101]. At 9% RH the effective sorption coefficient was found to be two-times higher than that of $S_{eff.}$ at 52% RH, which reflected the degressive increase of the water vapor sorption isotherm of silica gel (Figure 9). In comparison to the PE-LD, the effective sorption coefficients of the blends were by a factor of 200 to 1400 higher.

The water vapor absorption isotherm of silica gel in the filled films was calculated, using the water vapor absorption capacity in Table 4, and is compared with the water vapor absorption isotherm of pure silica gel in Figure 9. The water vapor absorption capacity of the silica gel in the polymer matrix was up to two-thirds lower compared to pure silica gel. *Pehlivan* et. al. [31] observed a similar behavior for polypropylene (PP) with dispersed zeolites. The filled film absorbed a factor of 4 to 7 less water vapor than the pure zeolite, the authors did not investigate the reasons. Possible explanations for the effects in our study, as mentioned before, were the penetration of the silica gel within the polymer and the compression of silica gel during extrusion (see Section 3.1). The high melt pressure during extrusion (PE-LD: 13–18 bar; 0.2 g silica gel/g film: 18–23 bar; 0.4 g silica gel/g film: 50–60 bar; 0.6 g silica gel/g film: 70–100 bar) were favorable for both effects. Furthermore, immobilization of active sites in silica gel by volatile decomposition products during the extrusion were principally possible. *Pesaran* reports about a capacity loss of up to 70% due to the exposure to cigarette smoke and thermal cycling [102]. A further explanation could be hydrothermal degradation of silica gel during extrusion. We assumed that low traces of water in the silica gel due to its absorption of water vapor in the environment could have caused loss of capacity and change in its structure. *Bühlmann* reported that silica gel in contact with water vapor at increased temperatures of up to 200 °C (melt temperature in this study was up to 210 °C) caused decrease of surface area and increase of pore diameters (less smaller pores) in silica gel [103]. Hydrothermal changes of the silica gel structure, such as lower surface area and higher pore diameters already at 200 °C, were observed by *Leboda et al.* and *Charmas* [104,105,106,107]. 

### 3.5. (Effective) Water Vapor Permeation Coefficients

In Table 5 the effective permeation coefficients are shown that were calculated from the effective diffusion, (Table 3) and the effective sorption coefficients (Table 4), using Equation (6). The effective diffusion and sorption coefficients were experimentally measured. Both procedures resulted in similar results. Compared to PE-LD, the permeation coefficient of filled films increased approximately by a factor of 2, 5, and 14 when they are filled with 0.2, 0.4, and 0.6 g silica gel/g film, respectively.

The permeation coefficients obtained by means of permeation measurements differed less than 50% compared to the effective permeation coefficient values obtained by means of the sorption measurements. For the assessment of this result, it is relevant to consider that permeation coefficients are often depicted in logarithmic scales and the permeability of polymers can differ by several magnitudes of order [108]. A variability of the permeability by a factor of two is not uncommon for polymers of same specification or type. In another publication, it was shown that results of both methods differed by up to a factor of 3 (whey protein films), in few cases were up to a factor of 5 (lipid-based films) [109].

In models according to *Maxwell*, *Bruggeman* and *Higuchi*, the permeability reduces when the concentration of filler particles in the polymer increases [92,110,111,112]. The theories assume impermeable particles (spheres) in a polymer matrix [111,112,113,114]. However, an increase of the water vapor permeability was observed in materials made of polymers and zeolites [115,116,117,118]. This behavior was explained by defects around the filler particles as it is the case here. As mentioned before, we assumed that channels and surface diffusion was present in the polymer matrix (Figure 2) with silica gel, besides that silica gel is a nanoporous material. The hypothesis of ours is in agreement with other studies. It was shown by *Willet* that effective diffusion coefficients, measured by sorption experiments for PE films with 55 wt.-% dispersed starch increased by a factor of 4 to 9 compared to films with 11 to 44 wt.-% dispersed starch [119]. With a density of starch of 1.5 g/cm^3^ and a density of polyethylene of 0.94 g/cm^3^, 55 wt.-% corresponds to 43 vol.-% starch [120], which is in the range of the highly filled samples of this study. *Peanasky* reported that starch dispersed in polyethylene with more than 30 to 40 vol.-% formed continuous channels [121]. *Wu* examined the water permeation at stretched polyethylene films with dispersed calcium carbonate. At more than 40 wt.-% calcium carbonate in the polymer matrix, the water vapor permeation increased strongly (40 wt.-% calcium carbonate corresponded to circa 20 vol.-% at a density of calcium carbonate of 2.71 g/cm^3^ and a density of polyethylene of 0.94 g∙cm^−^^3^) [122]. Results from *Huang*, *Gonzo*, *Balköse,* and *Friess* are qualitatively in accordance to our results. There, the permeation coefficient increased by the dispersion of porous zeolites in polymers [115,116,117,118].

A hypothesis of ours was that the effective sorption coefficient of LD-PE with silica gel inversely scales the effective diffusion coefficient. To relate the results for the effective diffusion, sorption, and permeation coefficients, the effective diffusion and sorption coefficients were depicted as pairs of values (Figure 10). The transverse lines are values for constant effective permeation coefficients that were determined by permeation measurements. On these lines, the products of effective diffusion and effective sorption coefficients are constant ($P_{eff.}=S_{eff.}\times D_{eff.}$), resulting in the measured effective permeation coefficient. This kind of depiction was chosen before by *Kanehashi* and *Nagai* and *McCall* for different gases (carbon dioxide, nitrogen, and methane) and water [123,124]. The pair of values for the effective diffusion and the effective sorption coefficients lay on one transverse line, which represents the effective permeation coefficients determined by permeation measurements. Therefore, the sorption experiments resulted in similar results to the permeation measurement results, concerning the effective permeation coefficients. The effective sorption coefficient was getting smaller as the RH was increased. On the other hand, the values of the effective diffusion coefficients increased as the RH values increased. The variation of the values for the effective measured permeation coefficients was less than 30% from the average value. The effective sorption coefficient was indirectly proportional to the effective diffusion coefficient ($P_{eff.}\approx S_{eff.}\times D_{eff.};P_{eff.}\approx constant$). With this knowledge it is possible to estimate the effective diffusion coefficient when the effective permeation coefficient and the sorption coefficient or the absorption capacity, are already known. This is the most important result of this work.

### 3.6. Gas Transmission Rates

In Table 6, the permeation coefficients for nitrogen, oxygen, and carbon dioxide are shown. (The sample with 0.6 g silica gel/g film could not be measured.) The nitrogen (N_2_), oxygen (O_2_), and carbon dioxide (CO_2_) permeation coefficients decreased as the silica gel concentration increased, opposed to the results obtained for water vapor permeation. The water vapor permeation coefficient increased as the silica gel concentration increases (Table 5). This behavior can be explained by the free OH-groups of silica gel. PE-LD with dispersed silica gel had, therefore, a higher effective solubility for water vapor. The non-polar gases N_2_, O_2_, and CO_2_ do not interact with these polar groups and their diffusion in the polymer is blocked by the silica gel particles.

Our results are in agreement with the results of *Bendahou et al.*, who observed a higher water permeability, but a lower oxygen permeability at increasing zeolite concentrations [125]. *Wolińska-Grabczyk et. al.* have also observed similar results in mixed matrix membranes made of several polymers containing dispersed 13X zeolite. At higher zeolite concentrations, a lower CO_2_ and N_2_ permeability, but a higher water vapor permeability [126] was obtained. Films with 40 wt.-% zeolite had a 10% to 30% lower CO_2_- and N_2_-permeability compared to the value for the pure polymer. In the films with 40 wt.-% silica gel (this study) the effect was stronger. We reached a value for the CO_2_-, N_2_-permeability that was 75% lower compared to the value for pure PE-LD films. For water vapor the permeability in the study of *Wolińska-Grabczyk et. al.* reached a value of up to a factor of 2.7 higher compared to the value for the pure polymer [126]. For the films with 40 wt.-% silica gel of this study the water vapor permeability reached a value of a factor of 4.8 higher compared to the value for pure polymer. The measured permeation coefficient for oxygen of pure PE-LD was in the range of other reported values [108,127].

### 3.7. Mechanical Properties and Elongation

The tensile stress and strain at break are shown as a function of the silica gel concentration in Figure 11 and Figure 12.

The pure PE-LD-film had a similar tensile stress at break as the value reported by *van Krevelen* with 10 N/mm^2^ [128], it was lower than the value of 25 N/mm^2^ reported by *Bleisch* [127]. In the product data sheet for blown film (blowing ratio of 2:1), a tensile stress at break of 17 (transverse to machine direction) and 27 N/mm^2^ (parallel to machine direction) is reported [57]. The values measured in this study were lower, because the film was a cast film and not a biaxially oriented blown film.

The tensile strain at break, (l_break_-l_0)_/l_0_), where l represents the length, of pure PE-LD is about a factor of 3.2 lower than the value given by *van Krevelen*. *Bleisch* [127], who reported values in the range of 2 and 8. A tensile stress at break of 2 (parallel to machine direction) to 6 (transverse to machine direction) is reported in the product data sheet, which has these values because of the film orientation during blowing.

Dispersed silica gel in PE-LD caused a higher tensile stress at break, but the tensile strain at break was reduced, which indicated a more brittle material. The increase in the tensile strength is an indication for adhesion or interlocking between the filler and the polymer matrix [129]. A similar result was observed by *Kajtar et al.* for zeolite in polymer. At a concentration of up to ca. 45 vol.-%, tensile strength slightly increased [35]. Reduction of the tensile strain at break by fillers was observed for polyethylene with dispersed, non-surface treated CaCO_3_ [130], polyethylene with up to 55 wt.-% talc [131], PLA with up to 30 wt.-% talc [132], and zeolites [35] in PE-LD.

## 4. Conclusions

Silica gel is a desiccant that can be well dispersed in polymers such as PE-LD. From their calculated effective diffusion coefficients, *D_eff._*, the measured water vapor absorption capacity, *m_∞_*, and the density of the material, the absorption behavior of films at various thicknesses made thereof can be estimated.

The permeation coefficients and the effective diffusion coefficients were determined by two methods, sorption and permeation experiments. The additional benefits of these results can be summarized as follows: On the one hand, the effective sorption coefficient of filled polymers can be calculated when knowing the permeation coefficient and the effective diffusion coefficient from the transient state permeation measurement. On the other hand, the effective permeation coefficient of these films can be calculated from the transient sorption measurement. It has been shown that both methods provide comparable results.

For the application of dispersed silica gel in films, the increase both in brittleness and volume must be considered, which can reduce the dimensional stability. The water vapor permeation coefficient of PE-LD films filled with 0.2, 0.4, and 0.6 g silica gel/g film increased approximately by a factor of 2, 5, and 14 compared to non-filled PE-LD. The higher water vapor permeability of the filled polymer layers can lead to the requirement of an additional water vapor barrier in multilayer film structures.

Investigations for future studies are worthwhile to determine the reaction constant and reaction order of silica gel, as well as the Thiele modulus [133,134,135,136], to conclude if the system is reaction or diffusion limited. Furthermore, the reason why the measured density of the films was higher than the expected density needs to be investigated for possible explanations such as the penetration of the porous network of silica gel or potential decomposition of silica gel.

Our results are relevant for the development and design of desiccant and humidity absorbing materials, which are made of polymers with dispersed desiccants and absorbers.

## Figures and Tables

**Figure 1 materials-12-02304-f001:**
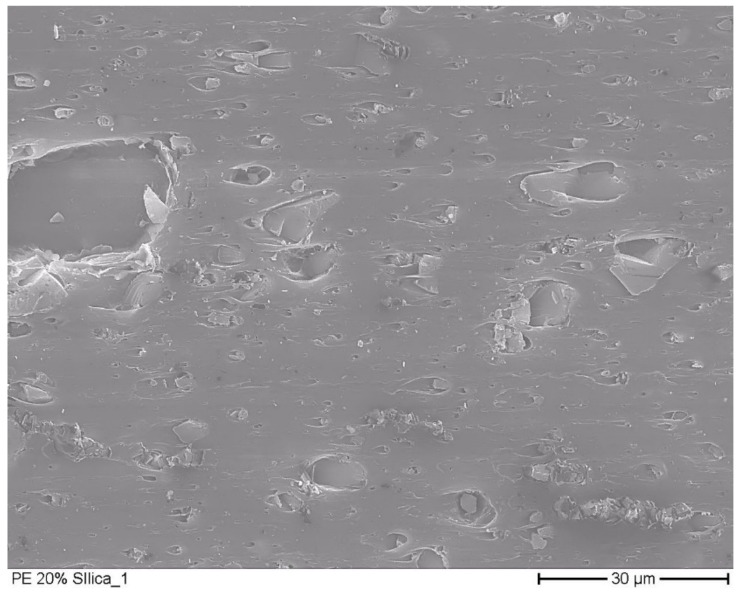
Electron microscopic picture of film with 0.2 g silica gel/g film.

**Figure 2 materials-12-02304-f002:**
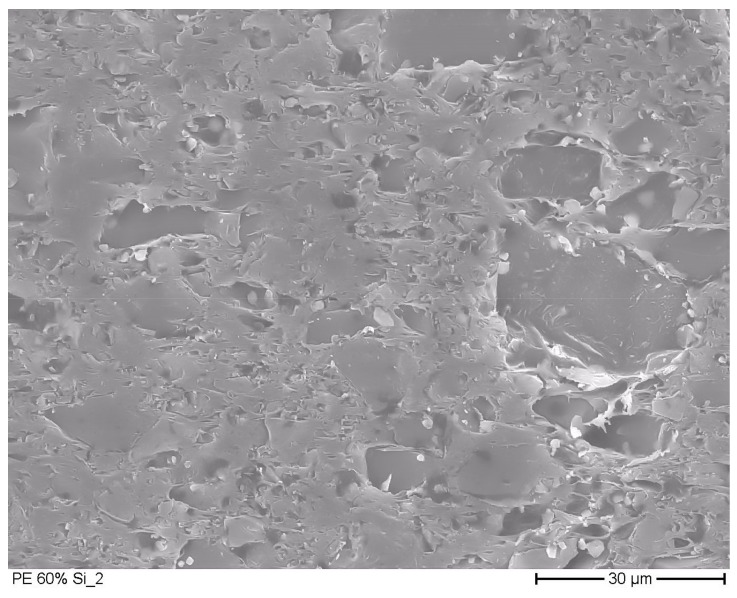
Electron microscopic picture of film with 0.6 g silica gel/g film.

**Figure 3 materials-12-02304-f003:**
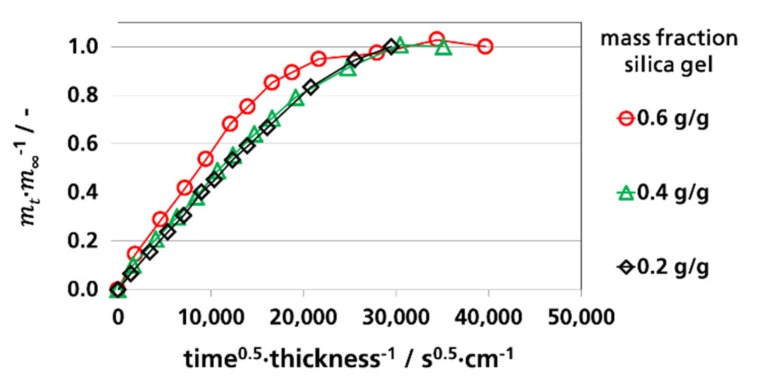
Water vapor absorption of films with dispersed silica gel at 9% RH and 23 °C, determination of $D_{eff.}$

**Figure 4 materials-12-02304-f004:**
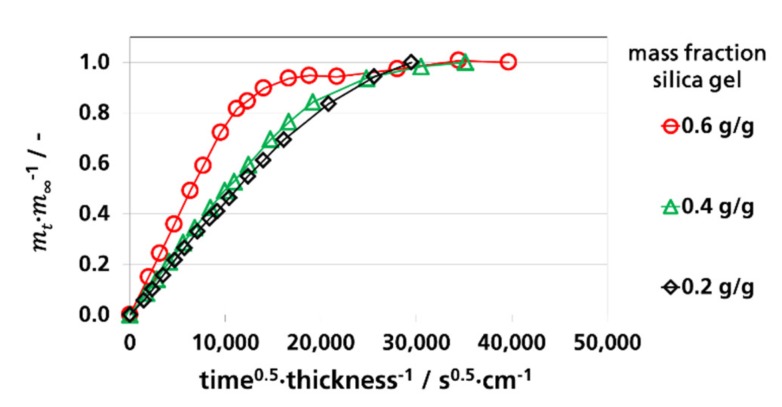
Water vapor absorption of films with dispersed silica gel at 52% RH and 23 °C, determination of $D_{eff.}$

**Figure 5 materials-12-02304-f005:**
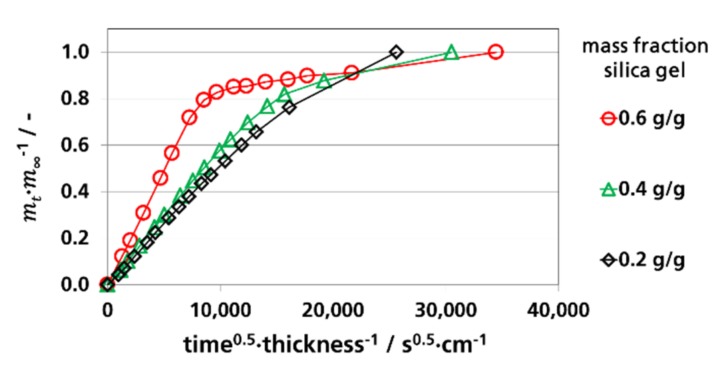
Water vapor absorption of films with dispersed silica gel at 91% RH and 23 °C, determination of $D_{eff.}$

**Figure 6 materials-12-02304-f006:**
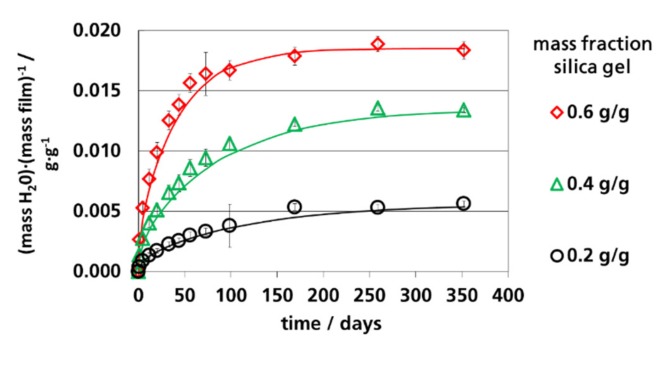
Water vapor absorption of films with dispersed silica gel at 9% RH and 23 °C, lines: Calculated using Equation (2) (*n* = 100).

**Figure 7 materials-12-02304-f007:**
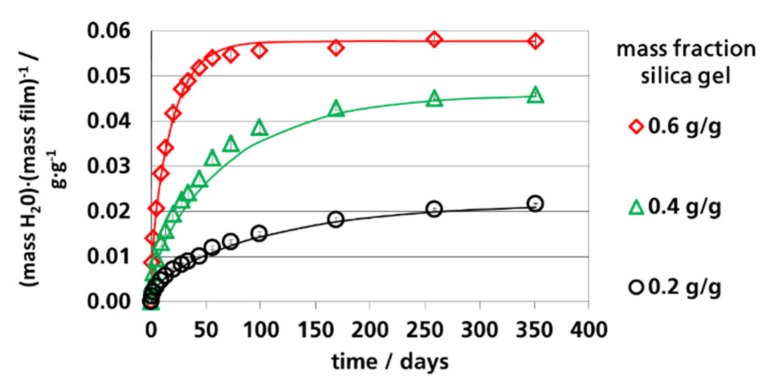
Water vapor absorption of films with dispersed silica gel at 52% RH and 23 °C, lines: Calculated using Equation (2) (*n* = 100).

**Figure 8 materials-12-02304-f008:**
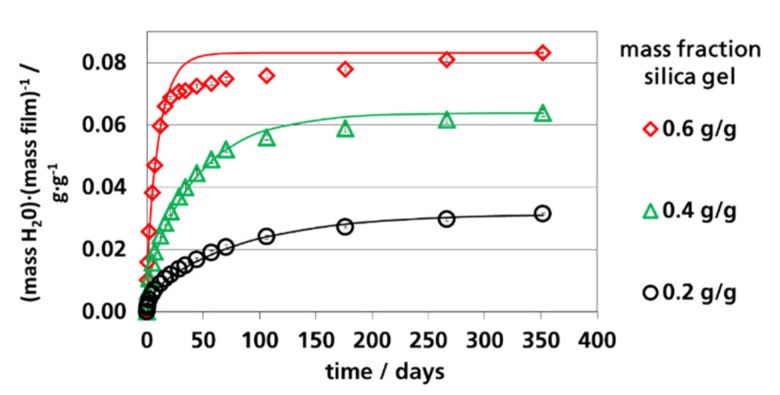
Water vapor absorption of films with dispersed silica gel at 91% RH and 23 °C, lines: Calculated using Equation (2) (*n* = 100).

**Figure 9 materials-12-02304-f009:**
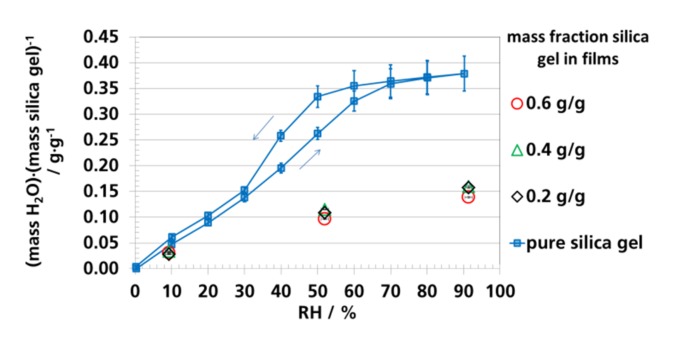
Water vapor sorption isotherm of pure silica gel and of silica gel in the polymer matrix (calculated from the mass fraction of silica gel in polymer and the measured absorption capacity of the films) at 23 °C.

**Figure 10 materials-12-02304-f010:**
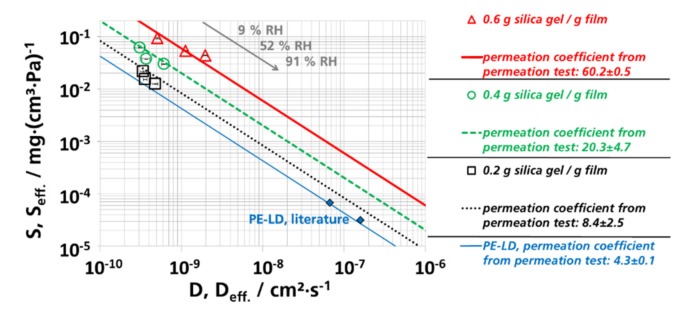
Pairs of values for the effective diffusion ($D_{eff.}$) and effective sorption coefficients ($S_{eff.}$ ), determined by sorption experiments; transverse lines represent permeation coefficients ($P_{eff.}$ ), determined by permeation measurements with the unit (mg·cm·(cm^2^·s·Pa)^−1^)·10^12^.

**Figure 11 materials-12-02304-f011:**
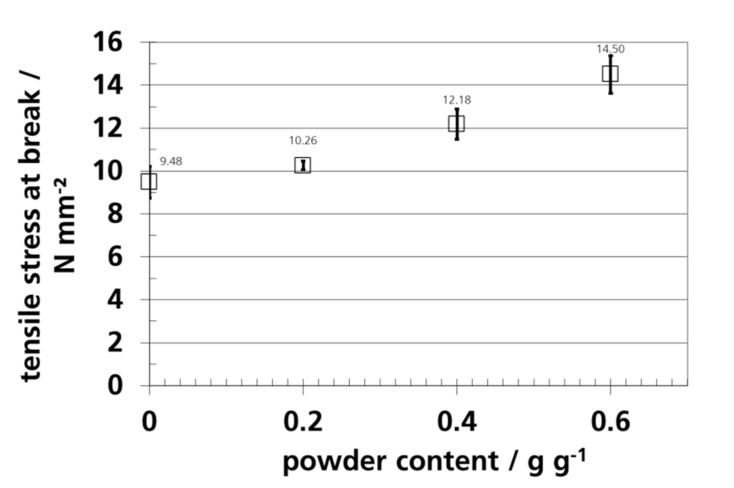
Tensile stress at break of PE-LD films with different amounts of silica gel powder.

**Figure 12 materials-12-02304-f012:**
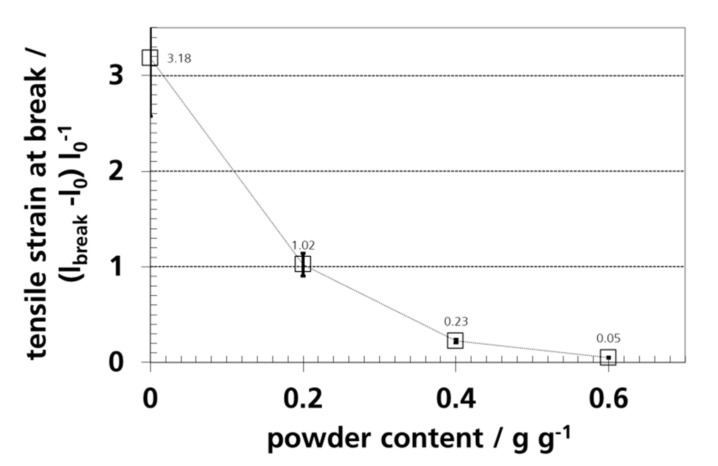
Tensile strain at break of PE-LD films with different amounts of silica gel powder.

**Table 1 materials-12-02304-t001:** Thicknesses of the films used for the sorption experiments, tensile tests, the determination of (effective) water vapor permeability coefficients, gas permeability.

Sample	Thickness/µm		
	Sorption Experiments (Extruded)	Tensile Tests (Extruded)	Permeability Tests (Extruded, Then Thermo-Pressed)
PE-LD	1754 ± 26	188 ± 27	315 ± 24
0.2 g silica gel/g film	1868 ± 28	1183 ± 29	357 ± 46
0.4 g silica gel/g film	1568 ± 37	768 ± 52	512 ± 72
0.6 g silica gel/g film	1391 ± 22	766 ± 79	232 ± 25

**Table 2 materials-12-02304-t002:** Measured and expected densities of films; * from [81].

Sample	Density/g·cm^−^^3^; Measured	Density/g·cm^−^^3^; Expected
0.2 g silica gel/g film	1.02 ± 0.01	0.99
0.4 g silica gel/g film	1.19 ± 0.01	1.04
0.6 g silica gel/g film	1.35 ± 0.01	1.10
comparison: PE-LD	0.94 ± 0.01	0.91, 0.94*
comparison: Silica gel	-	1.24
comparison: SiO_2_	-	2.2

**Table 3 materials-12-02304-t003:** Diffusion coefficient, *D*, of low-density polyethylene (PE-LD) [91] and effective diffusion coefficient, *D_eff._,* of filled film samples for water vapor determined by sorption (transient state) and by permeation (transient state) experiments (lag time determination by Equation (5)); at 23 °C; * from literature [91].

Sample	D, D_eff._/(cm^2^·s^−1^)·10^−10^				D_eff. Permeation_/D_eff. Absorption_
	From Sorption Test			From Permeation Test; 85 → 0% RH	
	9% RH	52% RH	91% RH		
Comparison: PE-LD	-	-	-	627, 482, 670*	-
0.2 g silica gel/g film	3.4 ± 0.3	3.6 ± 0.2	4.8 ± 0.4	5.0	1.4 to 1.5
0.4 g silica gel/g film	3.1 ± 0.4	3.7 ± 0.3	6.1 ± 0.4	4.7	0.8 to 1.5
0.6 g silica gel/g film	5.1 ± 0.3	11.3 ± 0.7	19.7 ± 1.1	12.2	0.6 to 2.4

**Table 4 materials-12-02304-t004:** Water vapor absorption capacity ($m_{\infty }$) of films; calculated sorption coefficient *S_eff._* (Equation (4)); at 23 °C; * sorption coefficient from published value [91].

Sample	$\;m_{\infty }\cdot 100/g\;H_{2}O\;(g\;\mathrm{Film}{)}^{-1}$		
	9% RH; 261 Pa	52% RH; 1460 Pa	91% RH; 2555 Pa
Comparison: PE-LD	-	~0	-
0.2 g silica gel/g film	0.56 ± 0.02	2.17 ± 0.02	3.14 ± 0.03
0.4 g silica gel/g film	1.34 ± 0.02	4.59 ± 0.05	6.39 ± 0.11
0.6 g silica gel/g film	1.84 ± 0.07	5.78 ± 0.11	8.32 ± 0.11
	**S, S_eff._/(mg H_2_O·(cm^3^ polymer·Pa)^−1^)·10^−2^**		
Comparison: PE-LD	-	0.0067*	-
0.2 g silica gel/g film	2.19 ± 0.09	1.52 ± 0.02	1.26 ± 0.02
0.4 g silica gel/g film	6.12 ± 0.09	3.76 ± 0.04	2.99 ± 0.05
0.6 g silica gel/g film	9.47 ± 0.28	5.33 ± 0.09	4.39 ± 0.05

**Table 5 materials-12-02304-t005:** Effective permeation coefficients calculated from results of sorption experiments and permeation coefficients measured with steady-state permeation experiments (steady state); 23 °C.

Film Samples	Permeation Measurements (mg·cm·(cm^2^·s·Pa)^−1^)·10^12^				Peff. Permeation Peff.Absorption^−1^-
	$\mathrm{Effective}\;\mathrm{Permeation}\;\mathrm{Coefficient}\;\mathrm{Obtained}\;\mathrm{by}\;\mathrm{Sorption}\;\mathrm{Test}\;(P_{eff.}\approx S_{eff.}\times D_{eff.})$			Permeation Test	
	9% RH	52% RH	91% RH	85 → 0% RH	
Comparison: PE-LD	-	-	-	4.26 ± 0.11	-
0.2 g silica gel/g film	7.4 ± 0.8	6.6 ± 2.9	6.0 ± 0.4	8.4 ± 2.5	1.1 to 1.4
0.4 g silica gel/g film	19.0 ± 2.6	13.9 ± 1.0	18.2 ± 1.2	20.3 ± 4.7	1.1 to 1.5
0.6 g silica gel/g film	48.0 ± 3.2	60.1 ± 4.8	86.4 ± 5.0	60.2 ± 0.5	0.7 to 1.3

**Table 6 materials-12-02304-t006:** Effective permeation coefficients calculated from results of sorption experiments in comparison to the measured steady state permeation coefficients at 23 °C; error bar: Maximal deviation from mean value.

Sample	Permeation Coefficients/(mg·cm·(cm^2^·s·Pa)^−1^)·10^14^, 23 °C			(Effective) Permeation Coefficients/(mg·cm·(cm^2^·s·Pa)^−1^)·10^12^
	N_2_	O_2_	CO_2_	H_2_O
PE-LD	6.3 ± 0.7	25.4 ± 2.7	110.6 ± 7.5	4.3 ± 0.1
0.2 g silica gel/g film	2.8	13.8	60.4	8.4 ± 2.5
0.4 g silica gel/g film	1.5 ± 0.2	7.8 ± 1.3	26.2 ± 7.4	20.3 ± 4.7

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
