# Peer review of "Desiccant Films Made of Low-Density Polyethylene with Dispersed Silica Gel—Water Vapor Absorption, Permeability (H2O, N2, O2, CO2), and Mechanical Properties"

_materials, 2019, doi:10.3390/ma12142304_

Round 1
Reviewer 1 Report
Comments to the Authors:
The work presented by the authors has the potential to be of interest to the scientific community and those who utilize desiccants to increase the shelf life of various consumer goods. Overall, the manuscript is well written and only minor typing and spelling errors were found and are noted below. A few major issues that should be addressed are:
1) In the introduction (line 51-65) the explanation of how silica gel works as a desiccant and its properties is very descriptive and relevant to the study; however, the connection/motivation between why silica gel was chosen over the other materials listed in the previous sentence is not clear. Connecting these dots will substantially increase the merit of the study presented.
2) Section 3.1. It appears that the conclusion was reached that the data presented in table 2 is not valid and the reasons why are merely speculated. This data should be removed. Since the nominal concentrations were only subsequently used, a simple statement, such as, “Attempts to determine the silica gel concentration in the films were unsuccessful and thus the nominal concentrations were used”, would be sufficient.
3) Section 3.2. No discussion on the formulation with 0.4 g of silica gel were made and should be included.
4) Section 3.4. One argument made for the lower water absorption of the films versus pure silica was due to the presence of decomposition products from the extrusion process. Are the degradation products coming from the LD-PE? The melting temperature for the LD-PE is listed as 108 ℃ in the experimental. At what temperature does the LD-PE degrade? Why isn’t the extrusion performed at a lower temperature if the polymer melts at 108 ℃?
5) Section 3.6. Can the permeation coefficients in table 6 be formatted to have the same units? It is difficult to follow the arguments made in the text (line 414-421). The coefficients for nitrogen, oxygen, and carbon dioxide decrease, but the way it is presented in the table it looks like they increase with respect to water because the units do not match.
6) Section 3.7. What was the motivation for the elongation study on a sample held at 91%RH for 260 days? It is not clear why this study was done. Without further explanation, Figure 14 and the small description (page 16, line 462-465) seem unimportant and should be removed.
Other minor issues are noted and should be corrected or addressed by the authors prior to accepting the manuscript.
In the title add “Gas” in front of Permeability and remove (H2O, N2, O2, CO2). The gases are listed in the abstract and the title is a bit wordy as is.
Page 1, line 18; H2O not H20 (zero was used instead of the letter O)
Page 1, line 20; PE-LD is never adequately defined. Consider rewording this sentence to : “Low density polyethylene (PE-LD) monolayer films with a nominal silica gel concentration of 0.2, 0.4 and 0.6 g 19 dispersed silica gel per 1 g film (PE-LD) were prepared and they absorbed up to 0.08 g water vapour 20 per 1 g of film.
Page 1, line 26; remove (effective).
Page 1, line 32-34; remove the last sentence of the abstract. Concluding remarks should be included in the manuscript and conclusion.
Page 1, line 43; change packagings to package
Page 2, line 46-47; This sentence is worded oddly. Consider changing to “Dispersing desiccants…”
Page 2, line 50; appears to be a double space between polymers and have.
Page 2, line 52; relative humidity is used here but in other instances RH is used. Be consistent and be sure to define RH at first mention.
Page 2, line 80-81; consider removing the sentence beginning “Aim of the milling…”. Not necessary to explain why something was done in this section, just how. The why should be addressed in the discussion section. Also, what size was the silica gel after ball milling?
Page 3, line 115; remove (effective)
Page 3, line 113; consider revising the figure heading to better note that the numbers in the image represent the dimensions of the mixing zones.
Page 3, lines 120-122; remove the first two sentences of this paragraph. The dimensions are included in Table 1 and it is not necessary to describe that thick ones were used for sorption and thin ones for tensile testing.
Page 4, line 129-131; the variables in the text vs equation 1 do not match. The variables need to be better defined. For example “The mass fraction of silica gel, x, can be determined …” vs what is there now
Page 4, line 137; consider removing “to have samples with the required electrical conductivity and electron density for the analysis.”
Page 5, line 180-181; a different font was used here.
Page 6, line 200-201; specimen should be plural
Page 6, line 206; missing parenthesis
Page 6, line 220; What is Annex 1?
Page 11, line 317; the table has the values at 0.6 to 2.4 not 0.62 to 2.39.
Page 12, line 334-335; missing period after reasons.
Page 12, line 349; formatting of some of the borders of the table are off. Should the S, Seff value for PE-LD be 0.0067 not 0,0067?
Page 14 line 389; change to “A hypothesis of ours”
Page 14, line 414-415; Why couldn’t the 0.6 g silica gel sample be measured for gas transmission?
Page 15, line 435-437; formatting of some of the borders of the table are a bit off.
Author Response
See file.

Reviewer 2 Report
This article by Sängerlaub et al.describes interesting results polyethylene-silica gel desiccant films. Though the concept of related materials may be well known, they investigated detailed synthesis, structure, and properties. Therefore, I think this manuscript can be published in Molecules.Please check the comments listed below and make a revision.
1. In order to mix inorganic and organic materials, we usually use silane coupling agents to avoid aggregation. Can you observe the particle size of silica in each film? If they are small enough compared to the one before mixing, this method is very effective.
2. Line 81, authors indicated that after milling silica gel became much smaller. If you have a data of particle diameter, please show.
3. Table 1: It seems that films containing more silica can be stronger and more permeable. However, that containing 0.6 g silica gel shows much less permeability than 0.4 g silica gel film. Please explain.
4. Table 2: Authors explain that the film that shows higher density than silica gel may be the results of compression of silica gel. Can you show the silica gel particle size before and after mixing?
5. Figure 11: There are only two spots for PE-LD value (in a literature), and silica films has three points. Is this graph really meaningful? It seems black dot line (0.2g silica gel) must show more steep line.
6. Other minor points:
Line 18: and after: H2O is shown in H20 (zero).
Line 206: extra ) after “Section 3.4”.
Line 360: extra space after “means of”
Author Response
Please see file attached.

Reviewer 3 Report
Dear Editor,
List of criticisms follows for the MS by Sängerlaub et al. “Desiccant Films Made of Low Density Polyethylene with Dispersed Silica Gel – Water Vapour Absorption, Permeability (H2O, N2, O2, CO2) and Mechanical Properties”.
The subject fits to the journal scope and, in my opinion, is interesting for the scientific community but the structure of the paper is practically identical to the paper:
Journal of Applied Polymer Science Volume 136, Issue 16 page 47460 “Desiccant films made of low-density polyethylene with dispersed calcium oxide: Water vapor absorption, permeation and mechanical properties”.
1) figure 1: this figure (caption included) is the same as in “Journal of Applied Polymer Science; Volume 136, Issue 16. A previous work published by the authors. (probably permission is needed)
2) SEM: Informations (WD, KV,…) should be added.
3) Many equations are the same as in the previous paper. Please check if permissions are needed.
What can I say about this work? Apart the same structure of a previous work, the Ms is well written and the investigations are well explained. Even the results are interesting.
The idea to add Silica Gel to this polymer seems to be a good idea.
The Authors probably have to fastly check with a software if some sentence is just copied and pasted from the previous work to this one by mistake.
I recommend the paper for publication after major revision.
Author Response
See file.

Round 2
Reviewer 3 Report
Dear Editor,
In the resubmitted manuscript the authors answered point by point to all of my considerations.
The authors spent time to answer in a proper way and enhanced the quality of their work.
This paper is recommended for publication.
Author Response
Thank you.